# Epigenetic-Like Stimulation of Receptor Expression in SSTR2 Transfected HEK293 Cells as a New Therapeutic Strategy

**DOI:** 10.3390/cancers14102513

**Published:** 2022-05-19

**Authors:** Joerg Kotzerke, Dorothee Buesser, Anne Naumann, Roswitha Runge, Lisa Huebinger, Andrea Kliewer, Robert Freudenberg, Claudia Brogsitter

**Affiliations:** 1Department of Nuclear Medicine, University Hospital Carl Gustav Carus, Technical University Dresden, Fetscherstrasse 74, D-01307 Dresden, Germany; dorothee.buesser@insel.ch (D.B.); anne.naumann@ezag.com (A.N.); roswitha.runge@uniklinikum-dresden.de (R.R.); lisa.huebinger@uniklinikum-dresden.de (L.H.); robert.freudenberg@uniklinikum-dresden.de (R.F.); claudia.brogsitter@uniklinikum-dresden.de (C.B.); 2Department of Pharmacology and Toxicology, Jena University Hospital, Friedrich Schiller University Jena, Drackendorfer Straße 1, D-07747 Jena, Germany; andrea.kliewer@med.uni-jena.de

**Keywords:** neuroendocrine tumors, peptide receptor radionuclide therapy, somatostatin receptor type 2, SSTR2, Lu-177-DOTATATE, valproic acid, 5-aza-2′-deoxycytidine

## Abstract

**Simple Summary:**

Neuroendocrine tumors (NETs) expressing the somatostatin receptor subtype 2 (SSTR2) are promising targets for peptide receptor radionuclide therapy (PRRT) using the somatostatin analogue Lu-177-DOTATATE. Patients expressing low levels of SSTR2 do not benefit from PRRT. Therefore, an approach to increase the efficacy of PRRT utilizing the effects of 5-aza-2′-deoxycytidine (5-aza-dC) and valproic acid (VPA) on the SSTR2 expression levels is investigated. The cell lines HEKsst_2_ and PC3 are incubated with 5-aza-dC and VPA in different combinations. The drug pretreatment of HEKsst_2_ cells leads to increased Lu-177-DOTATATE uptake values (factor 28) and lower cell survival (factor 4) in comparison to unstimulated cells; in PC3 cells, the effects are negligible. Further, for the stimulated cell types, the maintenance of the intrinsic radiosensitivity in each cell type is confirmed by X-ray irradiation. The increased SSTR2 expression induced by VPA and 5-aza-dC stimulation in HEKsst_2_ cells might improve treatment strategies for patients with NETs.

**Abstract:**

The aim of the study was to increase the uptake of the SSTR2-targeted radioligand Lu-177-DOTATATE using the DNA methyltransferase inhibitor (DNMTi) 5-aza-2′-deoxycytidine (5-aza-dC) and the histone deacetylase inhibitor (HDACi) valproic acid (VPA). The HEKsst_2_ and PC3 cells were incubated with variable concentrations of 5-aza-dC and VPA to investigate the uptake of Lu-177-DOTATATE. Cell survival, subsequent to external X-rays (0.6 or 1.2 Gy) and a 24 h incubation with 57.5 or 136 kBq/mL Lu-177-DOTATATE, was investigated via colony formation assay to examine the effect of the epidrugs. In the case of stimulated HEKsst_2_ cells, the uptake of Lu-177-DOTATATE increased by a factor of 28 in comparison to the unstimulated cells. Further, stimulated HEKsst_2_ cells demonstrated lower survival fractions (factor 4). The survival fractions of the PC3 cells remained almost unchanged. VPA and 5-aza-dC did not induce changes to the intrinsic radiosensitivity of the cells after X-ray irradiation. Clear stimulatory effects on HEKsst_2_ cells were demonstrated by increased cell uptake of the radioligand and enhanced SST2 receptor quantity. In conclusion, the investigated approach is suitable to stimulate the somatostatin receptor expression and thus the uptake of Lu-177-DOTATATE, enabling a more efficient treatment for patients with poor response to peptide radionuclide therapy (PRRT).

## 1. Introduction

Somatostatin (SST) is a peptide hormone that is released during digestion and takes part in the signal transduction of the initiation of apoptosis [1]. It is a crucial regulator of both the hormone and the nervous system and acts by binding to a G-protein-coupled receptor on the surface of various cell types. Out of the five subtypes of this receptor, the second type (SSTR2) is primarily addressed by synthetic lanreotide. Neuroendocrine tumors (NETs) demonstrate a heightened somatostatin receptor expression (SSTR), which can be addressed by a corresponding radioactive ligand in diagnostics and therapeutics [2].

A new approach does not focus on optimizing the ligand [3] but the target or rather the epigenetic-like stimulation of the extent of SSTR expressions [4,5]. Epigenetical changes, such as DNA methylation or histone deacetylation, can switch on tumor-suppressing genes or oncogenes, by chromatin condensation or hypermethylation in the DNA promoter region, for example. DNA methyltransferase inhibitors (DNMTi) and histone deacetylation inhibitors (HDACi) lead to remodeling of heterochromatin to euchromatin; the transcription factors are activated and epigenetically deactivated tumor-suppressing genes can be reactivated. Cell cycle control, double-strand break indication and apoptosis are initiated by 5-aza-dc [6]. Two substances of these so-called epidrugs are the DNMTi 5-aza-2′-deoxycytidin (5-aza-dC) and the HDACi valproic acid (VPA), which were used for our study. Several clinical studies investigated the application of 5-aza-dC alone or in combination with other anticancer agents [7]. Furthermore, 5-aza-dC is approved in clinical treatment of the myelodysplastic syndrome (Decitabine) [8]. VPA has been suggested as a radiosensitizer for a variety of cancers [9] and was used in a clinical trial involving patients with pancreatic cancer [10].

The epigenetical regulation of NET relies on a heightened *SSTR2* mRNA expression that modifies the protein expression of SSTR [11]. Individual observations reported that the SSTR expression can be increased by chemotherapy [12,13]. In addition, Cherk et al., as well as Adant et al. [14,15], recommend the increase of SSTR expressions and give an overview of previous in vitro, pre-clinical and in vivo applications in a review on combinational therapy by improving the effectiveness of peptide receptor radiotherapy.

After the initial study by Shimura et al. [16] regarding the gene transfer of the sodium iodine symporter (NIS) to non-iodine-avide thyroid carcinoma cells, Haberkorn et al. [17,18] transfected the NIS in hepatocellular and prostate carcinoma cells and could prove a substantially raised uptake of I-131 even in these cells. However, no intracellular retention of the radionuclide was achieved. Thus, the NIS became less important in therapeutical means and is mainly used as a reporter gene nowadays [19].

Furthermore, different subtypes of the somatostatin receptors have been cloned [20,21,22] and used later on as reporter genes [23,24].

In that, it is to be investigated if the SST2 expression in transfected HEK293 cells can be upregulated by epigenetic substances—analogous to primarily SST2-expressing cells—so that it can be addressed therapeutically [4,5].

The main goals of the investigation were to stimulate the SSTR2 expressions of the HEKsst_2_ and PC3 cells with the well-characterized epigenetically-efficient substances 5-aza-dC and VPA in combination, to prove the increased receptor expression by an increased uptake of Lu-177-DOTATATE. We examined a decreased cell survival in case of a Lu-177-DOTATATE treatment in comparison to unstimulated cells. In addition, we aimed at the analysis of intrinsic radiosensitivity potentially induced by the epidrug pretreatment.

## 2. Materials and Methods

### 2.1. Cell Culture

HEK293 cells were obtained from the German Resource Center for Biological Material (DSMZ, Braunschweig, Germany) and grown in Dulbecco’s modified Eagle´s medium supplemented with 10% fetal calf serum (FCS) in a humidified atmosphere containing 5% CO_2_. Cells were transfected with plasmid, encoding murine HA-tagged SSTR2 receptors using lipofectamine according to the manufacturing instructions (Thermofisher, Darmstadt, Germany). Stable transfectants were selected in the presence of 500 µg/mL G-418 and characterized using radioligand binding assays, Western blot analyses, surface ELISA assays and immunocytochemistry as described previously [25,26,27]. Non-transfected HEK293 cells were maintained under equal conditions as HEKsst_2_ except for the addition of G-418. In the following, HEK293 cells and SSTR2 transfected HEK293 cells are termed as HEK and HEKsst_2_, respectively. The human prostate carcinoma cell line PC3 (ATCC^®^, Manassas, VA, USA) was cultured in a RPMI medium with 10% (*v*/*v*) FCS and 1% (*v*/*v*) non-essential amino acids (NEA). The PC3 cell line exhibits a minimal expression of SSTR2 on the cell surface [28]. The culture medium of each cell line was renewed every 2–3 days. Via a PCR test, each cell line is proven to be free of mycoplasmas.

### 2.2. Chemical Modulation by 5-aza-dC and VPA

For all stimulation experiments, the DNA methyltransferase inhibitor (DNMTi) 5-aza-2′-deoxycytidin (5-aza-dC) and the histone deacetylation inhibitor (HDACi) valproic acid sodium salt (VPA) obtained from Merck KGaA, Darmstadt, Germany were used. To start, 5-aza-dC or VPA were dissolved in distilled water at stock solutions of 0.1 mM or 0.6 M, respectively. The chosen concentrations for experiments were prepared with dilutions in cell culture medium under sterile conditions.

### 2.3. Chemotoxicity of 5-aza-dC and VPA

The IC_50_ concentrations of 5-aza-dC and VPA as single modulators were analyzed for HEKsst_2_ and PC3 cells using a colony formation assay (CFA). HEKsst_2_ and PC3 were plated in T75 flasks on day zero. On day one, 5-aza-dC or VPA was added in the concentration regimes of 0–25 µM 5-aza-dC and 0–50 mM VPA. Cells were incubated in DMEM (HEKsst_2_) or an RPMI 1640 (PC3) nutrient-deficient medium (lacking medium additives and FCS) until day three at 37 °C. For a further two days, HEKsst_2_ and PC3 were incubated in a full medium (with medium additives and FCS). On day five, cells were trypsinized and plated for the CFAs.

### 2.4. Immunohistochemistry

The immunohistochemical (IHC) staining was performed to verify the SSTR2 expressions in the cell lines HEK and HEKsst_2_ with and without stimulation of epidrugs. The following combinations of 5-aza-dC/VPA concentrations were used, respectively: 0 µM/1.85 mM, 0.1 µM/1.85 mM, 3.9 µM/1.85 mM and 5.0 µM/1.85 mM. Each sample of HEK and HEKsst_2_ cells was transferred into cell blocks. Cell blocks were cut into 1 µm sections, and automatically stained with hematoxylin-eosin staining (Dako Omnis, Aligent Technologies, Santa Clara, CA, USA). SSTR2 IHC staining was performed using the monoclonal antibody anti-SSTR2A (Zytomed Systems GmbH, Bargteheide, Germany) in a dilution of 1:25. The detection was realized by an OptiView DAB Kit (760–700, Roche Holding AG, Basel, Switzerland) using the secondary antibody cocktail, goat-anti-mouse and goat-anti-rabbit, following the manufacturer’s instructions. All above-described immunostainings were processed with the Benchmark ULTRA IHC/ISH System (Roche). Cell sections were covered by an automated cover-slipper (Klinipath, Duiven, Netherlands), digitalized and analyzed by the case viewer software 2.4 (Sysmex GmbH, Norderstedt, Germany). Positive staining of SSTR2 expressions was defined as a brown staining pattern of cell membranes.

### 2.5. Irradiation, Radiosynthesis and Dosimetry

The radionuclide Lu-177 (LuCl_3_, non-carrier added, specific activity of 4110 GBq/mg) was provided by the company ITM GmbH (Isotopen Technologies, Muenchen, Germany). The radioactive labelling of (DOTA)-[Tyr3, Thr8]-octreotide (DOTATATE; ABX, Radeberg, Germany) was performed in a reaction buffer (sodium acetate, gentisic acid) with 10 µg DOTATATE and 500 MBq Lu-177. The radiochemical product purity was determined with ITLC (Instant Thin Layer Chromatography) and was ≥95% for all syntheses.

For the irradiation at the OncoRay site (National Center for Radiation Research in Oncology, Medical faculty Dresden), an X-ray tube (Y.TU 320, Yxlon International, Germany) with 200 kV X-rays (20 mA, dose rate ≈ 1.24 Gy/min, filtered with 0.5 mm Cu) was used.

To compare the applied X-ray radiation to the radioactivity, the activity of Lu-177 corresponding to the equivalent dose of X-rays is calculated. The calculation is done with Geant4 simulations for a 10 µm cell monolayer at the bottom of the well (9.6 cm²) in 2 mL cell culture medium. Following this model, only the extra cellular irradiation of the medium but not the dose contributions due to the cell-bound Lu-177-DOTATATE being accounted for [29]. The cell compartment-bound dose (membrane, cytoplasm and nucleus) can result in a specific dose increase. Due to cellular processes, such as the receptor-mediated binding of Lu-177-DOTATATE to the cell membrane, internalization, as well as externalization and recycling, whose detailed time courses were not analyzed, the results are displayed with respect to the radioactivity concentration of Lu-177-DOTATATE.

### 2.6. Treatment Design for Immunostaining

To study the effect of 5-aza-dC and VPA on SSTR2-expressions, HEKsst_2_ and HEK cells were plated in T75 flasks and both epidrugs were added in combination on day one. A drug-supplemented deficient medium was changed on day three; thereafter, cells were incubated for another two days as described in Section 2.3. On day five, cells were detached for immunostaining and plated in six-well plates for uptake studies. Four conditions were investigated in combined treatment using 5-aza-dC and VPA, as outlined in Section 2.4.

### 2.7. Measurement of Intracellular Lu-177-DOTATATE Radionuclide Uptake after Stimulation

To monitor the enhanced SST2 receptor expression, the internalized fraction of Lu-177-DOTATATE was measured after the five-day pre-incubation with the described concentrations of 5-aza-dC and VPA: 0 µM/1.85 mM, 0.1 µM/1.85 mM, 3.9 µM/1.85 mM and 5.0 µM/1.85 mM, respectively, including the samples without the addition of 5-aza-dC or VPA (unstimulated samples). After the stimulation process, 2.5 × 10^5^ cells in concentrations as specified in Section 2.4. were distributed as triplicates in 6-well-MTPs for uptake measurements using a full medium (without 5-aza-dC or VPA). The uptake measurements were performed with 50 kBq Lu-177-DOTATATE at incubations times of either 1 h or 24 h at 37 °C.

After incubation times, the supernatant of the well was removed and the cells were washed twice with PBS (phosphate buffered saline) at 4 °C. Supernatant and washed solution make up the non-bound fraction of the radioligand. The cell layers were lysed with 1 mL 0.1 M NaOH for 10 min at 37 °C. These fractions correspond to the intracellular fraction of radioligand. The activity of all fractions was measured using a Gamma-Counter (Cobra II, Autogamma, Packard^®^, Canberra Company, Ruesselsheim, Germany). The intracellular activity was expressed as a percentage of the total activity normalized to the cell count of 2.0 × 10^5^.

For saturation binding studies, stimulated HEKsst_2_ cells were incubated for 1 h at 37 °C with varying amounts of Lu-177-DOTATATE (3.5 × 10^−5^–7.03 × 10^−2^ nmol). Excess DOTATATE (3.5 × 10^−3^–7.03 nmol) was used to block receptors and determine the non-specific binding. The uptake data (triplicates per each condition) were fitted to a one-site binding model in GraphPad Prism 9.0 to determine the number of binding sites (B_max_) and the dissociation constant (K_d_).

### 2.8. Radiosensitivity and Radiotoxicity

To show the dose-response relationship, the unstimulated HEKsst_2_ and PC3 cells were irradiated with absorbed doses of 0.1–7.5 Gy in an X-ray tube. The dose-response of Lu-177-DOTATATE was observed after incubating HEKsst_2_ and PC3 cells with activity concentrations of 0.05, 0.1, 0.25, 0.5, 1.25, 2.5 and 5 MBq/mL over 24 h.

Considering the cytotoxicity of 5-aza-dC and VPA for the combined treatment of both cell lines, the IC_20_ and IC_40_ as well as the D_80_/A_80_ and D_60_/A_60_ for X-ray radiation and Lu-177-DOTATATE were used.

To examine the inherent radiation sensitivity of the stimulated cells, the 5-aza-dC and VPA concentrations and the absorbed doses of the X-rays as shown in Table 1 were used.

For the analysis of the effect of Lu-177-DOTATATE on stimulated HEKsst_2_ and PC3 cells, the equivalent radioactivity concentrations of 57.5 or 136 kBq Lu-177-DOTATATE/mL with an irradiation time of 24 h was used (Table 1). The stimulation process followed the protocol described above.

To determine the radiosensitivity and radiotoxicity, the clonogene cell survival was analyzed. Colony formation assays were performed as described earlier [30]. The following steps have been done for each of the treatment procedures. Briefly, after detaching the treated cells, an aliquot of the cell suspension per dose point was distributed in T25 culture flasks for the colony formation test and is placed in an incubator for 7 (HEKsst_2_) or 10 days (PC3). To stop colony formation, the cells were fixed with ethanol 80% (*v*/*v*) followed by staining with crystal violet solution. The colony counting was carried out under a microscope (25× magnification). Colonies of more than 50 cells were scored as survivors. The plating efficiency (PE) and the survival rate (survival fraction SF) were calculated for irradiated and unirradiated cells based on the number of seeded cells [31].

### 2.9. Statistical Analysis

All results of the uptake measurements and colony formation assay show the average values as well as the SEM (standard error of the mean) or SD (standard deviation) based on three independent trials, in which each experimental condition was assumed a triplicate. To examine the statistical significance, an unpaired student’s *t*-test was used. A difference between two independent samples is significant if the probability of error *p* ≤ 0.05. For the statistical evaluation, MS Office Excel was used.

The software Origin^®^Pro 2016 was used to determine the IC_50_, IC_40_ and IC_20_ concentrations of 5-aza-dC and VPA, respectively. To determine the B_max_ values and the K_d_, the GraphPad Prism 9.0 software was used. Confidence Intervals (CI) express a 95% likelihood of the nonlinear fit.

## 3. Results

### 3.1. Chemotoxicity of 5-aza-dC and VPA

Cell lines HEKsst_2_ and PC3 were incubated with varying concentrations of the epidrugs to determine the cytotoxic influence of both 5-aza-dC and VPA. The fraction of surviving cells (SF) are determined via CFA.

In the case of HEKsst_2_ cells, the half maximum inhibitor concentrations (IC_50_) of 5-aza-dC or VPA as single epidrugs are 3.9 µM 5-aza-dC and 1.85 mM VPA; for PC3 cells, the IC_50_ values are 0.3 µM 5-aza-dC and 3.9 mM VPA (Table 2). Based on these findings for the stimulation experiments of immunohistochemical staining, radioligand uptake and saturation binding assay, the following concentrations of 5-aza-dC/VPA were chosen, respectively: 0 µM/1.85 mM, 0.1 µM/1.85 mM, 3.9 µM/1.85 mM and 5.0 µM/1.85 mM. Assuming an additive effect due to combination of X-ray or Lu-177-DOTATATE with 5-aza-dC and VPA, the D_80_/A_80_/IC_20_ and D_60_/A_60_/IC_40_ were chosen to achieve SF values of approximately 0.6 or 0.35 (Table 2, Section 3.4 and Section 3.5)

### 3.2. Immunohistochemical Staining of Stimulated HEKsst_2_ and HEK Cells

HEKsst_2_ and HEK cells were incubated with a constant concentration of VPA according to the stimulation procedure described above. All concentrations of the stimulants are equal to those in Section 2.4. The immunohistochemical staining of the SST2 receptor showed a dominant expression of the membrane-bound receptor for HEKsst_2_ cells after treatment with 5-aza-dC and VPA. In comparison to HEKsst_2_ cells, for HEK cells without SST2 receptor, almost no SST2 receptor staining could be detected (Figure 1).

### 3.3. Uptake and Saturation Binding Assay in Dependency on 5-aza-dC and VPA Stimulation

When comparing cells with a stimulation of 5.0 µM 5-aza-dC and 1.85 mM VPA to unstimulated cells, a significant increase of Lu-177-DOTATATE uptake by a factor of 28 (50 kBq, 50.2% vs. 1.8%) is present after 1 h (*p* = 0.0142). After 24 h, an increase by a factor of 6.5 (50 kBq, 36.2% vs. 5.6%) was calculated (*p* = 0.0090). However, stimulated PC3 cells only showed a marginal increase of the uptake values compared to unstimulated cells after 1 h (*p* = 0.4935)) and 24 h (*p* = 0.3694) (Figure 2).

Figure 2 shows the findings of epidrug stimulation on the radioligand uptake. Compared with unstimulated HEKsst_2_, a stimulation of the HEKsst_2_ cells with 3.9 µM or 5.0 µM 5-aza-dC (and 1.85 mM VPA) significantly increases the Lu-177-DOTATATE uptake after 1 h by 2160% or 2800% (*p* < 0.05). Whereas the concentration of 0.1 µM 5-aza-dc also shows stimulatory effects, no significant difference was found (*p* = 0.060). After 24 h, the uptake significantly increased by ≈ 640% at concentrations of 3.9 µM and 5.0 µM 5-aza-dC, each in combination with 1.85 mM VPA (*p* < 0.05). However, stimulated PC3 cells showed only a marginal increase in uptake values after 1 h and 24 h compared to unstimulated cells. These differences are not statistically significant (*p* > 0.10), indicating no stimulatory effects of epidrug pretreatment for PC3.

In saturation binding experiments, the stimulation effects of 5-aza-dc and VPA on the specific uptake levels of HEKsst_2_ cells were analyzed in comparison to HEKsst_2_ cells without stimulation (Figure 3). The Bmax value for unstimulated cells was 4.36 × 10^−10^ nmol (CI = 4.15 × 10^−10^ to 4.58 × 10^−10^ nmol), and in the case of stimulation Bmax was 1.32 × 10^−9^ nmol (CI = 1.08 × 10^−9^ to 1.62 × 10^−9^ nmol). The calculation of the binding sites per cell using the Bmax values revealed a three-fold lower number of binding sites per cell (2.63 × 10^5^) for unstimulated HEKsst_2_ cells compared to stimulated HEKsst_2_ cells (7.95 × 10^5^). The Kd values under these assay conditions were 2.10 nM and 4.88 nM for unstimulated versus stimulated HEKsst_2_ cells, respectively.

### 3.4. Radiotoxic Effect of an X-ray on Unstimulated and Stimulated Cells

Looking at the dose-response for X-ray irradiation of up to 7.5 Gy without epigenetical pretreatment of the cells, a linear-quadratic model (Origin) is used for HEKsst_2_ and PC3 cells leading to the following D_37_ values: D_37_(HEKsst_2_) = 2.14 and D_37_(PC3) = 2.31. Both cell lines show a similar radiation sensitivity regarding X-rays (Figure 4). The X-ray dose points corresponding to the D_80_ and D_60_ values have been calculated from this dose response curve.

To investigate the effects of the stimulation regarding the inherent radiation sensitivity of the cells, 0.25 µM 5-aza-dC and 0.6 mM VPA with 0.6 Gy X-rays, as well as 0.46 µM 5-aza-dC and 1.0 mM VPA with 1.2 Gy X-rays were combined. The concentration of the epidrugs as well as the dose values correspond to the IC_20_/D_80_ and IC_40_/D_60_ values, respectively, as shown in Table 1. In Figure 5, the effect of the stimulation on the inherent radiation sensitivity with external X-rays is illustrated.

Since the cell survival fraction of 5-aza-dC and VPA stimulated cells had already decreased, these values were standardized to the initial value (SF = 1) for a better comparison to the untreated cells. A cell survival of 0.62 was found for a combined cell treatment with 5-aza-dC_0.25µM_ and VPA_0.6mM_ with additional 0.6 Gy X-ray radiation. The corresponding value for unstimulated cells was also 0.62 revealing no significant difference (*p* = 0.9695). For a stimulation with elevated concentrations of 5-aza-dC_0.46µM_ and VPA_1.0mM_ and an exposure to 1.2 Gy X-rays, the survival fractions were 0.57 and 0.46 for unstimulated cells (*p* = 0.1561), respectively. The statistical analysis of the results shows that there is no significant influence of the epigenetical stimulation on the radiation sensitivity of the HEKsst_2_ cells.

The combined stimulation of PC3 cells with 5-aza-dC_0.25µM_ and VPA_0.6mM_ with additional 0.6 Gy X-rays results to a cell survival of 0.52. Unstimulated cells exposed to 0.6 Gy X-rays showed a survival fraction of 0.87 (*p* = 0.1319). The stimulation at a dose point of 0.6 Gy induced lower cell survival than in unstimulated cells. The pretreatment with 5-aza-dC_0.46µM_ and VPA_1.0mM_ as well as the exposure to 1.2 Gy resulted in a cell survival fraction of 0.73. For unstimulated cells with an irradiation of 1.2 Gy, a similar survival fraction is found (0.81; *p* = 0.7822). Thus, the influence of an epigenetical stimulation regarding the intrinsic radiation sensitivity of both HEKsst_2_ and PC3 cells can be excluded.

### 3.5. Radiotoxic Effects of Lu-177-DOTATATE on Unstimulated and Stimulated HEKsst_2_ and PC3 Cells

In another experimental setting, the effect of the radiotracer Lu-177-DOTATATE on the unstimulated and epigenetically stimulated cell lines HEKsst_2_ and PC3 was investigated.

Both for the HEKsst_2_ and PC3 cells, Lu-177-DOTATATE induced a dose dependent decrease of the SF values. For Lu-177-DOTATATE concentrations of up to 0.5 MBq/mL, a clear decline in the cell survival fraction of HEKsst_2_ cells is visible, whereas the decrease of the survival of PC3 cells is much less pronounced. As the HEKsst_2_ express the SST2 receptors, we found a strong dose-dependent decrease of the SF values until the receptors seemed to be saturated. For high activity concentrations, the curve progression is almost parallel for both cell lines. Thus, it could be proven that an increase of the Lu-177-DOTATATE activity does not show an appropriate increase in the radiotoxic effect (Figure 6).

The stimulation experiments investigated if an increased expression of the SST2 receptors resulted in an increased cytotoxicity of the radiopharmaceutical. After treating the cells with 5-aza-dC and VPA with the IC_20_/IC_40_ and an irradiation with Lu-177-DOTATATE (A_80_/A_60_) corresponding to the values shown in Table 1, the clonogenic cell survival was investigated.

A survival fraction of 0.27 was measured for cells simultaneously stimulated with 5-aza-dC_0.25µM_ and VPA_0.6mM_ and irradiated with 57.5 kBq/mL Lu-177-DOTATATE (Figure 7). The irradiation of non-stimulated HEKsst_2_ cells with 57.5 kBq/mL Lu-177-DOTATATE resulted in a cell survival of 0.54. Although the SF decreased at 57.5 kBq/mL, no significant difference was reached (*p* = 0.1424). The application of a higher activity concentration of 136 kBq/mL Lu-177-DOTATATE showed a cell survival of 0.42 for unstimulated cells. On the contrary, the survival of the HEKsst_2_ cells stimulated with 5-aza-dC_0.46µM_ and VPA_1.0mM_ is significantly lower by a factor of 4.2 (SF = 0.09; *p* = 0.0029).

The cell survival of stimulated PC3 cells showed higher SF fractions for both 5-aza-dC and VPA concentrations in combination with 57.5 kBq/mL Lu-177-DOTATATE. The difference from unstimulated PC3 is not statistically significant (*p* = 0.5482). For 136 kBq/mL, only a slight, statistically insignificant difference in the survival fractions of stimulated and unstimulated cells was determined (*p* = 0.5078). As opposed to HEKsst_2_ cells, PC3 cells with a lower SST2 expression showed no increase regarding the radiotoxic effects when stimulated with 5-aza-dC and VPA, but rather protective effects towards irradiation.

## 4. Discussion

Enhancing the SST2 receptor levels can be an option to improve the therapeutic efficacy of PRRT for NETs expressing low levels of SSTR2. Similar to our results, previous studies investigated the SST2 stimulation using different cell lines as well as various epidrugs. Veenstra et al. [5] could prove that SSTR positive cell lines BON-1 and QGP-1 stimulated with 5-aza-dC and VPA show an increase of *SST2* receptor mRNA and an uptake enlarged by factor 40 (100 nM 5-aza-dC and 2.5 mM VPA). Thus, the target for octreotide therapy has been enlarged, which is documented by an octreotide-guided inhibition of the Forskolin-stimulated cAMP production. Taelman et al. [4] investigated numerous DNMTi and HDACi, including 5-aza-dC and VPA on cell lines BON-1, GQP-1, PC3 and AR42J and could prove an increase in mRNA and uptake. An increased uptake of Ga-68-DOTATOC was also shown by epigenetically stimulated BON-1 tumor-bearing naked mice.

Our Lu-177-DOTATATE uptake data displays that a combination of 5-aza-dC and VPA showed the maximum stimulation (factor 28) of the SSTR2 expressions of HEKsst_2_ cells, which is comparable to the results for BON-1 [4,32]. However, over 24 h, the cellular uptake of the radioligand decreased slightly in comparison to the uptake values obtained after 1 h incubation time. The reasons for the decrease of the stimulatory effects could be the chemotoxicity of the substances, the radiotoxic effects of Lu-177-DOTATATE activities used for uptake experiments as well as the desensitization of the receptor signaling leading to the reduction of receptor response over time [25]. In contrast, in PC3 cells, combined epidrug pretreatment had no effects on the uptake. This indicates that no SST2 stimulation occurs. These results are in agreement with the findings of Taelman et al. [4]. Observations regarding time intervals longer than 24 h have not been evaluated by the studies mentioned above [4,5]. In all, comparing our results to Veenstra et al., who found up to 3820 % higher uptake values of the untreated control, the difference in uptake increase can be explained by different experimental settings. Additionally, there are differences regarding the cell types as well as different 5-aza-dC and VPA concentrations.

The cell line HEKsst_2_ is not endogenously SST2-positive but was transfected. Nonetheless, it still shows a stimulability via 5-aza-dC and VPA analogous to primary SST2 positive cell lines such as BON-1, GQP-1 PC3 and AR42J. Independent of SST2 over-expression, we could show an enhanced SST2 receptor expression after 5-aza-dC and VPA stimulation in HEKsst_2_ cells compared to unstimulated cells. That might suggest an increase in *SST2* gene regulation. Recent investigations with the somatostatin receptor 4 agonist Veldoretide imply a similar stimulating effect for HEKsst_2_ cells compared to BON-1 [33], so that transfected cells maintain their functionality of SSTR.

The radiosensitivity expressed as SF after external X-ray irradiation is similar for the cell lines HEKsst_2_ and PC3 (Figure 4). After stimulation, both cell lines do not show systematic changes of the SF values (Figure 5). One exception is the dose point at 0.6 Gy X-rays for not stimulated vs. stimulated PC3 cells. Possibly, due to the variation of the SF values between the three independent experiments, the SF values do not reach a significant difference (*p* = 0.1319), although there is a clear difference between the SF values (0.87 vs. 0.52). Whereas the radiosensitivity of non-stimulated HEKsst_2_ and PC3 cells to X-rays is very similar (Figure 4), we found a clear difference between the SF values for HEKsst_2_ compared to PC3 in the stimulation setting. One explanation might be that the stimulation process is differently tolerated by the cell types. However, Taelman et al. found a lower survival of 5-aza-dC treated cells for PC3 compared to HEKsst_2_ [4].

The survival curve of HEKsst_2_ cells after incubation with Lu-177-DOTATATE show a sharp decline at low Lu-177-DOTATATE activity concentrations. With increasing concentrations of the radioligand, the curve progress becomes flatter, resulting in a parallel shift with respect to the survival curve of the PC3 cell line (Figure 6). The course of the curve demonstrates that the dose applied to the HEKsst_2_ cells is composed of the receptor-bound radioactivity and the radioactivity in the cell medium. Based on these findings, we assume that at a radioactivity concentration of 0.5 MBq/mL Lu-177-DOTATATE (equal to approximately 0.34 ng/500,000 cells), all SST2 receptors are engaged by either the radioactive-labeled ligand or the unlabeled ligand. At 0.5 MBq/mL Lu-177-DOTATATE, almost the maximum total binding of 17 kBq/500,000 cells is reached, as shown in Figure 3 (B_max_ = 4.4 × 10^−10^ nmol/cell) corresponds to 15.5 kBq/500,000 cells). Thus, a further increase in radioactivity concentration does not increase the cellular uptake but increases the dose via radiation from the cell culture medium containing Lu-177-DOTATATE.

After stimulation of the HEKsst_2_ cells, the cell survival reduces from 0.54 to 0.27 and from 0.42 to 0.1 depending on the concentration of both substances after application of 57.5 or 136 kBq/mL Lu-177-DOTATATE, respectively (Figure 7a). Thus, the radiotoxic effects increased by factors of 2 and 4 when comparing unstimulated and stimulated HEKsst_2_, respectively.

Transferring this approach to clinical situations, not single tumors cells but cancer tissue (>0.5 cm) is to be assumed. Consequently, clinical application could show even more the resulting effects than the colony formation assays due to a higher cross-dose in tissue compared to cell monolayers.

Whereas PC3 cells serve as a negative control (with at most minimal SST2 expressions), all five subtypes of SSTR in the LNCaP cell line could be enhanced with 5-aza-dC and Trichostatin proven by increased *SST2* mRNA transcription [34]. Similar behavior was found by Gailhouse et al. for adenocarcinoma as well as in pancreas and hepatocellular carcinoma [35,36]. Torrisani et al. showed that in cell lines with respectively low SST2 expressions, a significantly higher expression of SST2 receptors is reached when combining 5-aza-dC and Trichostatin A [37]. This observation backs the hypothesis that for further tumor entities, the SSTR expression can be stimulated epigenetically and that this concept can be transferred to other receptors or surface markers [38,39]. In case of SSTR, negative cells or other types of tumors, the SST2 receptor could be transfected and further stimulated, as the experiments by Li et al. and Noss et al. suggest [38,39].

Further, Chie et al. validated the radiosensitization effect of VPA for fractionated external radiation [40] and Jin et al. [41] observed significant effects on radiosensitization for 5-fluorouracil in combination with decitabine or tacedinaline.

An overview of studies investigating the synergistic improvement of cancer therapy using HDACi in clinical trials is given by Hontecillas-Prieto et al. [42]. Specifically, a randomized phase II trial of I-131-metiodobenzylguanidin (MIBG) in combination with the HDACi vorinostat shows beneficial radiation sensitizing properties. This combination provides a new option for patients with relapsed or refractory neuroblastoma [43].

Finally, our study revealed only a moderate decrease of cell survival (factors 2 or 4) when comparing the SF of stimulated vs. unstimulated HEKsst_2_ cells. These results are somewhat unexpected because stimulation using both epidrugs induced a clear increase of cell uptake and an enhancement of the receptor molecules for stimulated HEKsst_2_ cells. On the other hand, the saturation of Lu-177-DOTATATE cell binding at low radioactivity (0.5 MBq/mL) could explain these findings.

Since the receptor expression increases, the target of medicamentous, cellular or radiopharmaceutical approaches can be enlarged. There is a considerable potential for further studies regarding this background. Analogous to gene transfer for the NIS, transfection of the SST2 receptor offers a new approach for guided radionuclide therapy [44,45,46]. The additional stimulation of receptor expression via epigenetically effective substances is proven, even though the exact mechanism is not yet clarified.

## 5. Conclusions

It could be shown that the receptor expressions of genetically modified cells can be stimulated with the same drugs in a comparable magnitude as the genuine SST2 expressing cell lines. The Lu-177-DOTATATE uptake experiments revealed an increased uptake by a factor of 28 for stimulated HEKsst_2_ cells compared to the unstimulated cells. The stimulation did not affect the intrinsic radiosensitivity of the cells. Thus, the efficiency of the radionuclide therapy in vitro is enhanced by a factor of 4, whereas in vivo this factor should be even larger due to the higher cross dose from tissue composition.

Analogous to the transfection of NIS, the transfection of SST2 receptors depicts a new approach for gene therapy and is especially appealing since receptor expressions can be stimulated even more after transfection.

## Figures and Tables

**Figure 1 cancers-14-02513-f001:**
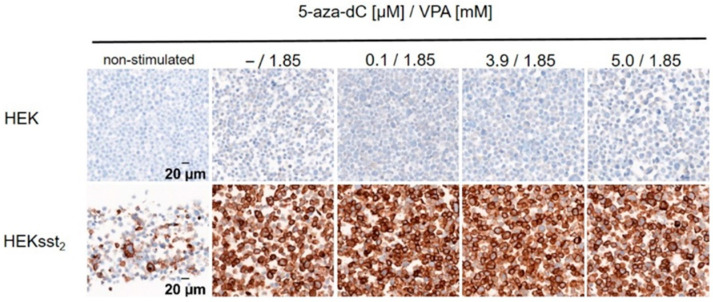
Representative image of SST2 receptor staining for HEK and HEKsst_2_ cells after stimulation by increasing concentrations of 5-aza-dC and constant concentrations of VPA. Non-stimulated HEKsst_2_ and HEK cells are shown as a reference (40× magnification).

**Figure 2 cancers-14-02513-f002:**
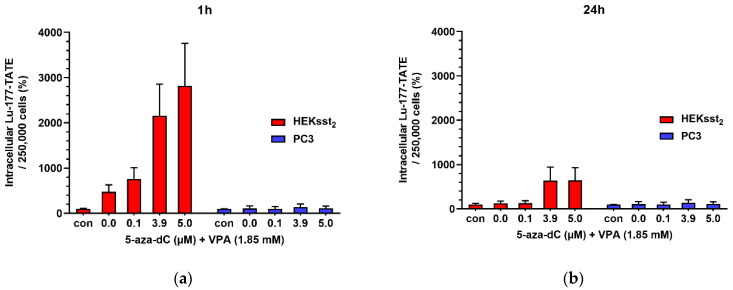
(**a**) Cellular uptake of Lu-177-DOTATE (50 kBq) for 1 h in HEKsst_2_ and PC3 cells and (**b**) 24 h. Before uptake analysis both cell lines were pretreated with 5-aza-dC and VPA over a time period of 5 days as described above. Unstimulated controls (con) were simultaneously incubated with Lu-177-DOTATATE. The intracellular activity is normalized to results in unstimulated control cells. Results are expressed as changes in percentage in relation to results without stimulation. All data are shown as mean ± SD.

**Figure 3 cancers-14-02513-f003:**
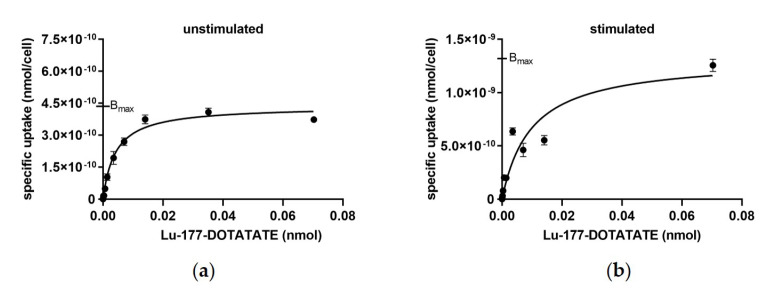
(**a**) Saturation binding curves of unstimulated HEKsst_2_ cells and (**b**) stimulated HEKsst_2_ cells. The non-specific binding, determined by blocking with excess of DOTATATE, was subtracted and only specific binding is shown. Data are expressed as average ± SD.

**Figure 4 cancers-14-02513-f004:**
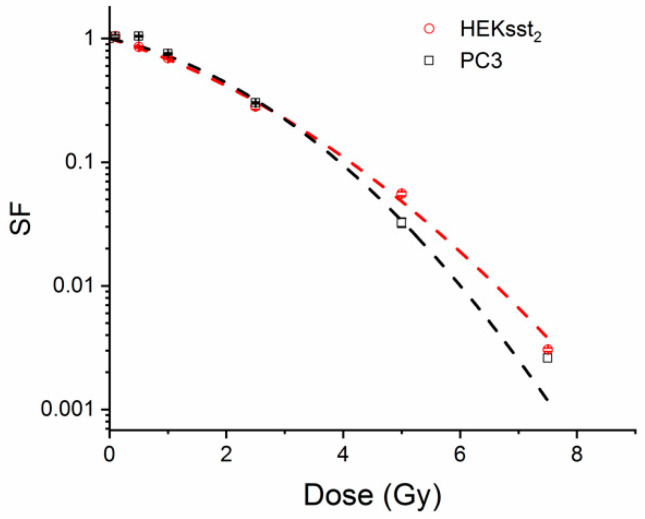
Cell survival curves for unstimulated HEKsst_2_ and PC3 cells after X-ray irradiation. Data are expressed as average ± SEM.

**Figure 5 cancers-14-02513-f005:**
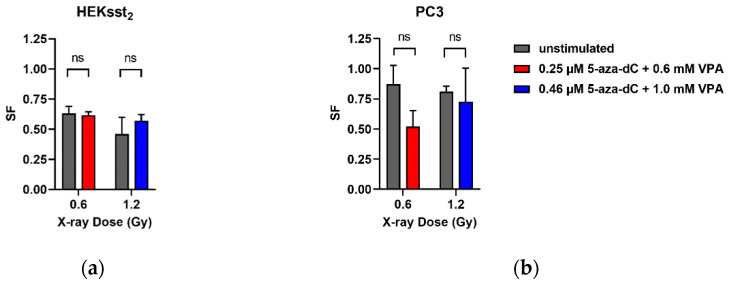
(**a**) Cell survival fractions for HEKsst_2_ and (**b**) PC3 cells after stimulation with 5-aza-dC and VPA and X-rays. Data are displayed as average ± SEM. ns: not significant.

**Figure 6 cancers-14-02513-f006:**
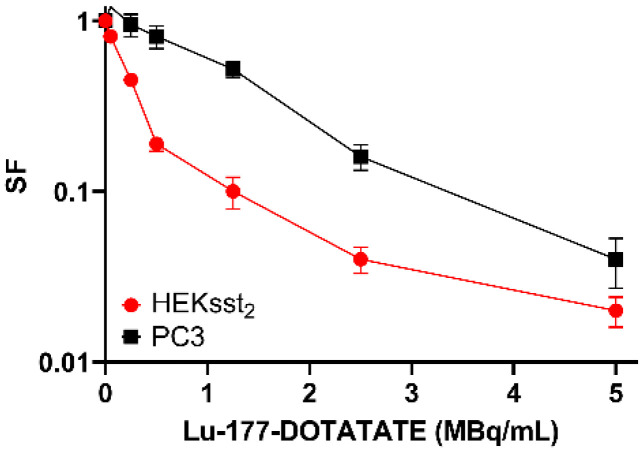
Radiotoxic effect of Lu-177-DOTATATE on unstimulated HEKsst_2_ and PC3 cells. The averages ± SEM are indicated.

**Figure 7 cancers-14-02513-f007:**
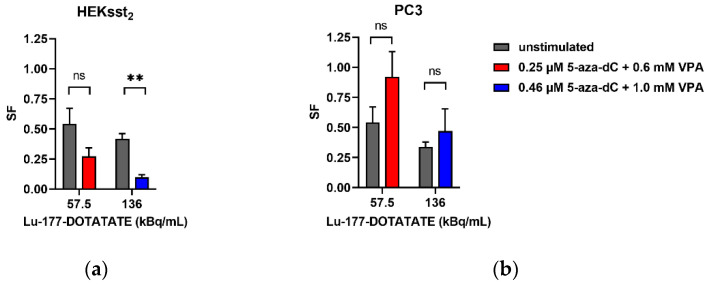
(**a**) Cell survival fractions of HEKsst_2_ and (**b**) PC3 cells after combined treatment with 5-aza-dC and VPA and irradiation with Lu-177-DOTATATE (average ± SEM). The SF values of stimulated cells were normalized to SF = 1 according to Figure 5. ns: not significant, ** *p* < 0.01.

**Table 1 cancers-14-02513-t001:** IC_20_/D_80_/A_80_ and IC_40_/D_60_/A_60_ values for cell lines HEKsst_2_ and PC3 used for radiosensitivity and radiotoxicity experiments.

Treatment Conditions	IC_20_/D_80_/A_80_	IC_40_/D_60_/A_60_
5-aza-dC (µM)	0.25	0.46
VPA (mM)	0.6	1.0
X-ray (Gy)	0.6	1.2
Lu-177-DOTATATE (kBq/mL)	57.5	136

**Table 2 cancers-14-02513-t002:** Chemotoxicity of 5-aza-dC and VPA for cell lines HEKsst_2_ and PC3 presented as survival fractions (%).

	Concentration 5-aza-dC (µM)								
Survival Fraction (%) ± SD	0	0.01	0.05	0.1	0.5	1.0	5.0	10	25
HEKsst_2_	100 ± 6.2	97.9 ± 1.8	118.0 ± 31	83.2 ± 4.8	96.6 ± 13.2	93.7 ± 1.3	42.3 ± 4.8	18.6 ± 0.8	1.7 ± 0.4
PC3	100 ± 3.6	67.1 ± 3.3	52.1 ± 19.0	69.4 ± 2.1	41.6 ± 1.8	24.9 ± 2.0	27.5 ± 3.2	9.4 ± 0.3	1.9 ± 0.02
	Concentration VPA (mM)								
Survival fraction (%) ± SD	0	0.1	0.5	1.0	2.5	5.0	10	25	50
HEKsst_2_	100 ± 10.7	183.1 ± 7.2	78.7 ± 10.8	67.5 ± 15.9	18.5 ± 2.2	21.0 ± 5.1	53.5 ± 5.8	8.0 ± 0.8	n.a. ^1^
PC3	100 ± 8.1	95.6 ± 4.5	95.6 ± 2.3	72.6 ± 11.6	69.4 ± 6.0	35.1 ± 4.6	24.0 ± 1.2	1.2 ± 1.1	1.7 ± 1.1

^1^ not applicable.

## Data Availability

Data can be requested by contacting the corresponding author.

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
