# Peer review of "Epigenetic-Like Stimulation of Receptor Expression in SSTR2 Transfected HEK293 Cells as a New Therapeutic Strategy"

_cancers, 2022, doi:10.3390/cancers14102513_

Round 1

Reviewer 1 Report

NET tumors can be successfully treated with peptide radionuclide therapy (PRRT) using the somatostatin analog Lu-177-DOTATATE only in patients expressing the somatostatin receptor. The main aim of the work was to increase the expression of this receptor through the combined use of epigenetically efficient substances 5-aza-dC and VPA in PC3 and HEKsst2 cells and to demonstrate decreased cell survival in case of a treatment with Lu-177-DOTATATE compared to non-stimulated cells. In addition, the authors studied the intrinsic radiosensitivity potentially induced by epidrug pretreatment.

I personally consider the work very beautiful and interesting. The authors have studied several interconnected endpoints and have shown that they have a clear experimental design.

Here are some suggestions and requests for clarification.

I would add in the last paragraph of the introduction the reason why they were chosen precisely 5-aza-dC and VPA as stimulating agents instead of others. Alternatively, perhaps it is sufficient to add a reference to line 81 after the sentence “ ….known epigenetically-efficient substances 5-aza-dC and

VPA……”.

I found inconsistencies in the materials and methods and in the results and some things are not at all clear to me.

Chemotoxicity tests (paragraph 3.2) led to the identification of IC50 and from this, the authors chose a series of combined concentrations. These are the same as shown in Figure 1 relating to the immohistochemical staining of the receptor SST2. Is not it more logical and clear to show the toxicity data first and then the receptor expression?

Also in paragraph 3.1, the authors write “HEKsst2 and HEK cells were incubated with varying concentrations of 5-aza-dC and VPA according to the stimulation procedure described above”. This sentence is not correct as the concentration of VPA remains constant at 1.85 mM and varies only that of 5 -aza-AD. Furthermore, reference is made to table 1 (“All concentrations of the stimulants are equal to those in Table 1”) which, on the other hand, seems to me to report data for other endpoints studied.

Could the authors clarify the expression “a single noxa ….”? Do you mean a single experiment? Is this why the error in Table 2 is shown as standard deviation instead of standard error of the mean? Also, could the authors clarify the expression “noxa experiments”? Do you mean a single experiment? Furthermore, some errors seem particularly large reaching values of around 24% and 26% of the measured value. However, do the authors believe the data to be reliable considering that the IC20 and IC40 used for the evaluation of radiotoxicity are also obtained from this curve?

Some thoughts about Figure 5. Since SF is plotted, why didn't the authors use the semilog scale? Moreover, in my opinion, being plotted only two dose points the representation with histograms would have been more correct and less misleading. The fit shown does not make much sense with only two points !!!!.

I ask also to the authors to verify the point at 0.6 Gy of PC3 not stimulated and treated with 5-aza-dC 0.25 μM and VPA 0.6 mM. A non-significant p-value ((p = 0.9695) is indicated which would seem to be incorrect, considering that the error bars do not touch and the two points are quite different from each other. The section of the discussion related to these results, should be modified accordingly.

In Figures 6 and 7, the authors show the radiotoxic effect of Lu-177-DOTATATE in unstimulated and stimulated cells, respectively. In the case of unstimulated cells, the SF decrease is observed in both lines with a steeper and more linear part up to 0.5 MBq / mL for HEKsst2 cells. In my opinion the sentence  “For a Lu-177-DOTATATE concentration of 0.5 MBq / mL a clear decline in the cell  survival fraction of the HEKsst2 cells is visible” should be rewritten because the clear decline is observable for both cell lines. I am not clear even the sentence “The linear display in the diagram inset represents the effect additionally” and the inserted graph is very small and not very readable. Maybe it should be better explained what the authors mean by “effect addionally”. Regarding Figures 7a and 7b the same considerations made for Figures 5a and 5b apply: graph the data as a histogram, do not indicate the fit and check for significance.

The discussion is overall well-structured and supported by other scientific work on the topic. In my opinion, some points, however, are in need of improvement. First, the authors write: "However, cellular uptake was not stable for 24 hours, but showed a time-dependent decreasing retention". This does not appear when looking Figures 2a and 2b. In the results the same authors state that after 24h there is an increase of a factor 6.5 in stimulated HEKsst2 while PC3 only showed marginal increase of the uptake values for both times studied. Clarify this point.

Last but not least the minor corrections:

Line 106: to cancel “were used” because is repeat at the end of the sentence

Line 171: delete “(paragraph 2.5)” and replace it with the correct one (2.3 or 2.6 I don't know)

Line 207: authors should verify the accuracy of the table header. Are they sure they are dose response curves without stimulation? The presence of the stimulants would make me think otherwise

Line 240: delete “and VPA” because the its concentration is not increased

Line 248 e 242: delete the S. This is Table 2. Change chemotixicity in chemotoxicity

Line 290: delete Table 2 and write Table 1

Line 282 and 284: delete the “S” in Figure 4

Line 379: use SF

Line 380: delete the “S” in Figure 4

Author Response

 Dear Reviewer 1, thank you very much for your extensive review and the detailed, helpful remarks. We considered each point carefully and made corrections, additions or clarified statements.

Reviewer 1

Comments and Suggestions for Authors

NET tumors can be successfully treated with peptide radionuclide therapy (PRRT) using the somatostatin analog Lu-177-DOTATATE only in patients expressing the somatostatin receptor. The main aim of the work was to increase the expression of this receptor through the combined use of epigenetically efficient substances 5-aza-dC and VPA in PC3 and HEKsst2 cells and to demonstrate decreased cell survival in case of a treatment with Lu-177-DOTATATE compared to non-stimulated cells. In addition, the authors studied the intrinsic radiosensitivity potentially induced by epidrug pretreatment.

I personally consider the work very beautiful and interesting. The authors have studied several interconnected endpoints and have shown that they have a clear experimental design.

Here are some suggestions and requests for clarification.

  1. I would add in the last paragraph of the introduction the reason why they were chosen precisely 5-aza-dC and VPA as stimulating agents instead of others. Alternatively, perhaps it is sufficient to add a reference to line 81 after the sentence “ ….known epigenetically-efficient substances 5-aza-dC and VPA……”.

Changes on this point were made on page 2. 

  1. I found inconsistencies in the materials and methods and in the results and some things are not at all clear to me. Chemotoxicity tests (paragraph 3.2) led to the identification of IC50 and from this, the authors chose a series of combined concentrations. These are the same as shown in Figure 1 relating to the immohistochemical staining of the receptor SST2. Is not it more logical and clear to show the toxicity data first and then the receptor expression?

We agree to this comment. The paragraph order has been changed.

  1. Also in paragraph 3.1, the authors write “HEKsst2 and HEK cells were incubated with varying concentrations of 5-aza-dC and VPA according to the stimulation procedure described above”. This sentence is not correct as the concentration of VPA remains constant at 1.85 mM and varies only that of 5 -aza-AD. This sentence has been corrected.

  1. Furthermore, reference is made to table 1 (“All concentrations of the stimulants are equal to those in Table 1”) which, on the other hand, seems to me to report data for other endpoints studied.

Information to all concentrations is corrected in the manuscript. The reviewer is right, we used different concentration ranges in terms of the endpoints.

  1. Could the authors clarify the expression “a single noxa ….”?

The expression “a single noxa” means that the cells have been treated with 5-aza-dC or VPA alone, respectively. Unfortunately, we can’t find the expression “noxa experiments” in the manuscript.

  1. Do you mean a single experiment? Is this why the error in Table 2 is shown as standard deviation instead of standard error of the mean? Do you mean a single experiment?

For chemotoxicity experiments 2-3 independent experiments with each point in triplicates were performed.

  1. Also, could the authors clarify the expression “noxa experiments”?

Unfortunately, we can’t find the expression “noxa experiments” in the manuscript.

  1. Furthermore, some errors seem particularly large reaching values of around 24% and 26% of the measured value. However, do the authors believe the data to be reliable considering that the IC20 and IC40 used for the evaluation of radiotoxicity are also obtained from this curve?

The authors agree, there are partly large errors. Errors occur at low concentrations of 5-aza-dC and VPA were not chosen for experiments.

Additionally, our setting followed the approach of the isobolographic analysis: Assuming an additive effect due to combination of X-ray or Lu-177-DOTATATE with 5-aza-dC and VPA the D80/A80/IC20 and D60/A60/IC40, were chosen to achieve SF values of approximately 0.6 or 0.35, respectively. Thus, the IC20 and IC40 values served rather as an estimation of concentration ranges to avoid very low survival rates and thereby, a high number of combination experiments. Thus, we are sure that our IC20 and IC40 are reliable for this setting.

  1. Some thoughts about Figure 5. Since SF is plotted, why didn't the authors use the semilog scale? Moreover, in my opinion, being plotted only two dose points the representation with histograms would have been more correct and less misleading. The fit shown does not make much sense with only two points !!!!.

Yes, that’s right, we absolutely agree. Please find the changed Figure 5 in the manuscript.

  1. I ask also to the authors to verify the point at 0.6 Gy of PC3 not stimulated and treated with 5-aza-dC 0.25 μM and VPA 0.6 mM. A non-significant p-value ((p = 0.9695) is indicated which would seem to be incorrect, considering that the error bars do not touch and the two points are quite different from each other. The section of the discussion related to these results, should be modified accordingly.

This is a regrettable mistake on ours part. The correct p-value (p= 0.1319) is now changed in the manuscript. The authors corrected the p-value in the results section as well as modified the discussion.

  1. In Figures 6 and 7, the authors show the radiotoxic effect of Lu-177-DOTATATE in unstimulated and stimulated cells, respectively. In the case of unstimulated cells, the SF decrease is observed in both lines with a steeper and more linear part up to 0.5 MBq / mL for HEKsst2 cells. In my opinion the sentence  “For a Lu-177-DOTATATE concentration of 0.5 MBq / mL a clear decline in the cell  survival fraction of the HEKsst2 cells is visible” should be rewritten because the clear decline is observable for both cell lines. I am not clear even the sentence “The linear display in the diagram inset represents the effect additionally” and the inserted graph is very small and not very readable. Maybe it should be better explained what the authors mean by “effect addionally”.

The sentence has been rewritten. Figure’s 6 inset is removed because the decline in the cell survival fraction is also observable in the semilogarithmic scale for HEKsst2 and PC3.

  1. Regarding Figures 7a and 7b the same considerations made for Figures 5a and 5b apply: graph the data as a histogram, do not indicate the fit and check for significance.

Figures 7a and 7b have been displayed by histogram plot now. Significances have been checked and marked in the graph.

The discussion is overall well-structured and supported by other scientific work on the topic. In my opinion, some points, however, are in need of improvement.

  1. First, the authors write: "However, cellular uptake was not stable for 24 hours, but showed a time-dependent decreasing retention". This does not appear when looking Figures 2a and 2b. In the results the same authors state that after 24h there is an increase of a factor 6.5 in stimulated HEKsst2 while PC3 only showed marginal increase of the uptake values for both times studied. Clarify this point.

In Figures 2a and 2b the data are displayed as changes in percentage in relation to results without stimulation. So, the difference in results between 1h and 24h becomes more clearly.

In the discussion section explanations have been made to the reviewer comment.

Last but not least the minor corrections:

Line 106: to cancel “were used” because is repeat at the end of the sentence It is done.

Line 171: delete “(paragraph 2.5)” and replace it with the correct one (2.3 or 2.6 I don't know)

It is replaced by 2.4.

Line 207: authors should verify the accuracy of the table header. Are they sure they are dose response curves without stimulation? The presence of the stimulants would make me think otherwise

The authors would say, that the D80/A80 D60/A60 calculations based on the dose response curves shown in Figures 4 and 6. In the table header this expression lead to misunderstanding.

Line 240: delete “and VPA” because the its concentration is not increased

It is done.

Line 248 e 242: delete the S. This is Table 2. Change chemotixicity in chemotoxicity

It is done.

Line 290: delete Table 2 and write Table 1

It is done.

Line 282 and 284: delete the “S” in Figure 4

It is done.

Line 379: use SF

It is done.

Line 380: delete the “S” in Figure 4

It is done.

Reviewer 2 Report

This manuscript described the possibility that the effect of  177Lu-DOTATATE therapy was enhanced by increasing the expression of SSRT2 by using DNMTi and HDACi. The concept of this study is interesting and would have an impact on PRRT. However, the proof of this concept from the results of this study seems to be weak since there is no in vivo study data which is the most important for evaluating the efficacy of the PRRT therapy. Thus, it should be better to reconstruct the design of this study and then revise this manuscript. In addition, I have some question and comment about this manuscript as written below.

  1. In page 2, line 80-84, "5-aza-dC" and "VPA" seem to appear suddenly. Thus, the detailed information should be added, which was already written in the materials and method section, and should be moved to the introduction section.
  2. In figure 1, the increased SSTR2 expression was saturated with 0.1 uM 5-aza-dC. On the other hand, the uptake levels of 177Lu-DOTATATE increased with the concentration of 5-aza-dC for 5.0 uM. Why did this discrepancy happen?
  3. In figure 2, the uptake levels of 177Lu-DOTATATE increased about 25 times by stimulating with 5-aza-dC + VPA, while the Bmax increased only twice by the stimulation. I think the Bmax is correlated with the expression levels of SSTR2, and thus uptake levels of 177Lu-DOTATATE should be correlated with Bmax.
  4. In the cytotoxic study, only PC3 cell was used as a negative control cell. I think HEK cell is preferred as a  negative control cell since the difference between HEKsst2 cells HEK cells is only the expression level of SSTR2.
  5. Why was the concentration of VPA different in the cellular uptake study of 177Lu-DOTATATE and cytotoxic study?
  6. Figure 7 seems to lead the audience to misunderstand, for example, the increase of 5-aza-dC and VPA improved the SF in figure 7a. The data should be displayed by a bar graph, not a scattered graph.
  7. In figure 6, the SF of HEKsst2 seems to be decreased significantly compared with that of PC3 at the point of 0.5  MBq/mL 177Lu-DOTATATE. On the other hand, in figure 7, it seems to be almost the same between HEKsst2 and PC3. Why does this discrepancy happen?
  8. The in vivo study should be performed as I mentioned at the beginning of my comment.
  9. In this study, the SSTR2 transfected model cell was used as a positive (SSTR2 high expression) cell. But, I wonder if the same phenomenon happens in the cells expressing SSTR2 highly and naturally (such as AR42J).

Author Response

Dear Reviewer 2, thank you very much for your helpful review and the detailed remarks. We considered each point carefully and made corrections, additions or clarified statements.

Comments and Suggestions for Authors

This manuscript described the possibility that the effect of  177Lu-DOTATATE therapy was enhanced by increasing the expression of SSRT2 by using DNMTi and HDACi. The concept of this study is interesting and would have an impact on PRRT. However, the proof of this concept from the results of this study seems to be weak since there is no in vivo study data which is the most important for evaluating the efficacy of the PRRT therapy. Thus, it should be better to reconstruct the design of this study and then revise this manuscript. In addition, I have some question and comment about this manuscript as written below.

1. In page 2, line 80-84, "5-aza-dC" and "VPA" seem to appear suddenly. Thus, the detailed information should be added, which was already written in the materials and method section, and should be moved to the introduction section.

Detailed information to 5-aza-dC and VPA is moved to the introduction section.

2. In figure 1, the increased SSTR2 expression was saturated with 0.1 uM 5-aza-dC. On the other hand, the uptake levels of 177Lu-DOTATATE increased with the concentration of 5-aza-dC for 5.0 uM. Why did this discrepancy happen?

This discreancy migt be explained by using to different methods that are different in terms of sensitivity and specifity. The immunohistochemical method uses SST2 antibodies for staining in fixed cells. In contrary, the uptake determines the integrity of the living cells and their receptors. The authors believe that this discrepancy might be related to these reasons.

3. In figure 2, the uptake levels of 177Lu-DOTATATE increased about 25 times by stimulating with 5-aza-dC + VPA, while the Bmax increased only twice by the stimulation. I think the Bmax is correlated with the expression levels of SSTR2, and thus uptake levels of 177Lu-DOTATATE should be correlated with Bmax.

The authors used two uptake approaches. For the measurement of intracellular Lu-177-DOTATATE we used a fix concentration of the radioligand (50 kBq) without incubation of cold ligand. Another setting was chosen for saturation binding experiments to determine the number of SST2 receptors per cell. Our experiments are performed using several concentrations of the radioligand, incubating each in the presence of 100-fold excess of cold ligand. The molecular principle underlying such experiments is that association of a radioligand with its receptor and dissociation of the ligand-receptor complex are reversible processes (law of mass action, non-linear). Receptor cycling, recycling and desensitization are processes that might be different in terms of the uptake approaches. A 3-fold higher number of receptors per cell after stimulation might be a comprehensible result.

4. In the cytotoxic study, only PC3 cell was used as a negative control cell. I think HEK cell is preferred as a  negative control cell since the difference between HEKsst2 cells HEK cells is only the expression level of SSTR2.

In principle the authors agree to this comment. We have discussed this point at the beginning of our experiments. For several reasons we decided to use not the HEK cells: Our intention has been the use of a cell line which is well established und frequently used. Further, the HEK cells are not easy to cultivate (primary cells, embryonic stem cells). They are semi-adherent to plastic materials, particular over the stimulation procedures. Also, the colony forming assays require stably adherent cells.

5. Why was the concentration of VPA different in the cellular uptake study of 177Lu-DOTATATE and cytotoxic study?

The concentration of VPA for the cellular uptake study is based on the results of chemotoxicity experiments (Table 2).

Our setting for radiotoxicity of stimulated cells followed the approach of the isobolographic analysis: Assuming an additive effect due to combination of X-ray or Lu-177-DOTATATE with 5-aza-dC and VPA the D80/A80/IC20 and D60/A60/IC40, were chosen to achieve SF values of approximately 0.6 or 0.35, respectively. Thus, the IC20 and IC40 values of 5-aza-dC and VPA were adjusted for radiotoxicity studies.

6. Figure 7 seems to lead the audience to misunderstand, for example, the increase of 5-aza-dC and VPA improved the SF in figure 7a. The data should be displayed by a bar graph, not a scattered graph. Figure 7 is displayed by a bar graph now, also figure 5.

7. In figure 6, the SF of HEKsst2 seems to be decreased significantly compared with that of PC3 at the point of 0.5  MBq/mL 177Lu-DOTATATE. On the other hand, in figure 7, it seems to be almost the same between HEKsst2 and PC3. Why does this discrepancy happen?

In figure 6 the radiotoxic effects on unstimulated HEKsst2 and PC3 are shown. Figure 7 shows the radiotoxic effects on stimulated cells.

8. The in vivo study should be performed as I mentioned at the beginning of my comment.

The authors agree, in vivo studies are required to transfer research findings into clinical applications. But, in vitro models are very helpful for research on basic molecular mechanism and may serve as an orientation for further in vivo studies. On the other hand, it is known, experiments on cell cultures are only partly applicable to living organism. An option might be the use of 3D cell experiments or studies using organoid test systems before animal testing is performed. Furthermore our infrastructure is not so well-developed for animal testing.

Altogether, we will take this comment as a suggestion for further studies.

9. In this study, the SSTR2 transfected model cell was used as a positive (SSTR2 high expression) cell. But, I wonder if the same phenomenon happens in the cells expressing SSTR2 highly and naturally (such as AR42J).

Taelman et al. demonstrated that DNMT and HDAC inhibition had no significant effects in AR42J but they used Decitabine (5-aza-dC) alone for stimulation no combination with another drug. Please find the reference in our manuscript.

Round 2

Reviewer 1 Report

I thank the authors for accepting the suggestions and making the required corrections. In my opinion, the article has now improved, more complete and suitable for publication.

A clarification. The authors responded to my request for an explanation of the term noxa but they could not find the term in the text. In the draft manuscript I received for revision from line 246 to line 248 the following sentence was written:

The half maximum inhibitor concentrations (IC50) of 5-aza-dC and VPA as single noxa are in the case of HEKSst2 cells 3.9 μM 5-aza-dC and 1.85 mM VPA; for PC3 cells the IC50 values are 0.3 μM 5-aza-dC and 3.9 mM VPA (Table S2).

In the revised version, the term is replaced by epidrugs.

There are some editing errors to be corrected:

line 177 (paragraph2.4.) a space is missing

line 190 250,000 perhaps it is preferable to write 2.0 x 105

line 254 0.35.Table 2 a space is missing. Missing title and caption of Table 2

line 294 stimulation. . All remove a dot

line 412 time.. remove a dot

line 413 effects could perhaps remove a space

Reviewer 2 Report

The authors revised their manuscript well in accordance with my comment. Thus, I would like to recommend it be published in Cancers.